# Comparison of Volatiles in Different Jasmine Tea Grade Samples Using Electronic Nose and Automatic Thermal Desorption-Gas Chromatography-Mass Spectrometry Followed by Multivariate Statistical Analysis

**DOI:** 10.3390/molecules25020380

**Published:** 2020-01-16

**Authors:** Shuyan Wang, Feng Zhao, Wenxi Wu, Pengjie Wang, Naixing Ye

**Affiliations:** 1College of Horticulture, Fujian Agriculture and Forestry University, Fuzhou, Fujian 350002, China; 1180311012@fafu.edu.cn (S.W.); wpjtea@163.com (P.W.); 2Key Laboratory of Tea Science in Fujian Provincial University, Fujian Agriculture and Forestry University, Fuzhou, Fujian 350002, China; 3College of Pharmacy, Fujian University of Traditional Chinese Medicine, Fuzhou, Fujian 350122, China; 4Hengzheng Testing Technology Co., Ltd., Fuzhou, Fujian 350100, China; wwx@hzhtt.com

**Keywords:** Chinese jasmine tea, tea grade, volatile organic compounds, electronic nose (E-nose), automatic thermal desorption-gas-chromatography- mass spectrometry (ATD-GC-MS), multivariate statistical analysis

## Abstract

Chinese jasmine tea is a type of flower-scented tea, which is produced by mixing green tea with the *Jasminum sambac* flower repeatedly. Both the total amount and composition of volatiles absorbed from the *Jasminum sambac* flower are mostly responsible for its sensory quality grade. This study aims to compare volatile organic compound (VOC) differences in authoritative jasmine tea grade samples. Automatic thermal desorption-gas-chromatography-mass spectrometry (ATD-GC-MS) and electronic nose (E-nose), followed by multivariate data analysis is conducted. Consequently, specific VOCs with a positive or negative correlation to the grades are screened out. Partial least squares-discriminant analysis (PLS-DA) and hierarchical cluster analysis (HCA) show a satisfactory discriminant effect on rank. It is intriguing to find that the E-nose is good at distinguishing the grade difference caused by VOC concentrations but is deficient in identifying essential aromas that attribute to the unique characteristics of excellent grade jasmine tea.

Chemical compounds studied in this article: methyl salicylate (PubChem CID: 4133), linalool (PubChem CID: 6549), linalool oxide (furanoid) (PubChem CID: 240), ethyl decanoate (PubChem CID: 8048)

## 1. Introduction

It is a general belief that the pleasant aroma of the *Jasminum sambac* flower can relieve the mood of depression [1]. Moreover, the health effect of tea also has been widely confirmed [2]. Both of these concepts make jasmine tea a popular tea worldwide [3,4]. Traditionally, the processing of jasmine tea, includes the following seven steps (shown in Appendix A) of tea dhool preprocessing, fresh flowers maintenance, tea and flower combination, scenting, flower removal, drying, and packing [5]. Current Chinese National Standards subdivide jasmine tea into six grades according to the number of times of repeated scenting which affects the quality of both the *Jasminum sambac* flower and the tea dhool [6,7]. The floral fragrance adsorption and persistence are critical factors related to jasmine tea grading [8].

Presently, there are existing studies on the evaluation of jasmine tea quality. Chen et al. observed the changes of volatile compounds during the scenting processes, and marked a serial of positive correlated compounds [3], for example. Lin et al. proposed a jasmine tea flavor (JTF) index (the ratio of peak area percentage of (Z)-3-hexenyl benzoate, α-farnesene, methyl anthranilate to linalool) as a novel quality evaluation index for jasmine tea’s volatile organic compound (VOC) evaluation [9]. Shen et al. believed the adsorption and retention of endogenous volatiles of tea was key for its quality [8]. Liang et al. analyzed the application of chemical composition and solution color to the difference of jasmine tea in its quality evaluation [10].

Electronic nose (E-nose) is another technique which has been widely used in product quality testing [11], medical diagnosis [12] and environmental monitoring [13], for example. E-nose can make a simple, fast and effective discrimination [14,15]. Its vital module is the sensor array of metal oxide films which can simulate the human nose and generate corresponding signals for gases. The response value of the e-nose is R/R0. R0 is the reference resistance obtained by cleaning the electronic nose before testing, and R is the sample resistance obtained during testing. While, E-nose is also a typical gray box system, which mainly constructs the discriminant model between input signals and output results through algorithm training [11], it means that, although the correct judgment could be given, it is still hard to tell which substances play a key role in grade contingencies.

Gas-chromatography- mass spectrometry (GC-MS), coupled with an enrichment pretreatment is the most commonly used method. Conventional existing enrichment methods include simultaneous distillation and extraction [16], headspace solid-phase microextraction [17], solid-phase extraction [18], accelerated solvent extraction [19] and more. Automatic thermal desorption (ATD) is a new prominent enrichment method, which has the advantages of convenient operation, a high enrichment rate, good reproducibility and no use of organic solvents. The combination of ATD to GC-MS has been used in air monitoring [20], analysis of pesticides in the atmosphere [21], material and emission analysis [22], food and aroma analysis [23] etcetera. The main advantage of applying ATD to the detection of jasmine aroma is the content of VOC enrichment could be much higher than that found by solid-phase microextraction (SPME).

The purpose of this study is to compare aroma characteristics within different jasmine tea grade samples through distinct techniques. A group of corresponding samples are subjected to research. Both electronic nose and automatic thermal desorption-gas-chromatography-mass spectrometry (ATD-GC-MS) are applied. Their discriminant effects are compared systematically. Accordingly, the rapid classification of jasmine tea is achieved using an electronic nose, while ATD-GC-MS detection followed by multivariate data analysis can provide a more profound understanding of the composition of volatile substances related to grading classification.

## 2. Materials and Methods

### 2.1. Sample Information 

A group of authoritative jasmine tea grade samples (including six grades, indicated as 1G, 2G, 2G, 3G, 4G, 5G and 6G, three repeats per grade for automatic thermal desorption-gas-chromatography-mass spectrometry (ATD-GC-MS) tests, six repeats per grade for the E-nose test) prepared according to Chinese National Standards GB/T 34779-2017 [5], were provided by Fujian Tea Import and Export Company Limited. (Fuzhou city, Fujian province, China). All samples were stored in a refrigerator at 4 °C before analysis.

### 2.2. Chemicals

Volatiles standards, including methyl salicylate (PubChem CID: 4133; 99.5%), linalool (PubChem CID: 6549; ≥99.5%), linalool oxide (furanoid) (PubChem CID: 240; ≥99.5%), ethyl decanoate (PubChem CID: 8048; ≥99%), were purchased from Aladdin (Shanghai, China).

### 2.3. Automatic Thermal Desorption-Gas-Chromatography-Mass Spectrometry (ATD-GC-MS) Analysis 

The volatile organic compound (VOC) of jasmine tea samples was analyzed using an ATD-GC-MS method, described by Zheng [24], with slight modification. A COLIN Tech Auto thermal desorption sampler (Chengdu Colin Analytical Technology Co., Ltd., Chengdu, China) and a Shimadzu 2010 gas-chromatography (GC) coupled with 8040 triple quadrupole mass spectrometry (TQ-MS) (Shimadzu Production Institute, Kyoto, Japan) was applied.

#### 2.3.1. Extraction of Volatile Organic Compounds (VOC)

A QC-1S atmosphere sampling instrument (Beijing Kean Labor Insurance New Technology Co., Ltd., Beijing, China) was used for VOC extraction according to China’s National Environmental Protection Standards [25]. The VOC analysis method was the same as Zheng et al. [24]. Briefly, 3.0 g of sample was weighted into a headspace bottle and ethyl decanoate (100 ppm, 15 µL) was added to the samples as the internal standard. Then, the headspace bottle was sealed and equilibrated at 55 °C for 20 min. Afterward, the sorbent tube (Chengdu Colin Analytical Technology Co., Ltd., Chengdu, China) was connected to the atmosphere sampling instrument and headspace bottle according to the flow direction of the sorbent tube with polytetrafluoroethylene (PTFE) pipes. Finally, volatile components were collected at 200 mL/min flow rate for 30 min. After sample collection, both ends of the sorbent tube were sealed with PTFE caps and transported to the laboratory for analysis.

#### 2.3.2. Thermal Desorption 

Thermal desorption was conducted by a COLIN Tech Auto thermal desorption sampler (Chengdu Colin Analytical Technology Co., Ltd., Chengdu, China). The primary thermal desorption of sampling tube was carried out at 250 °C for 5 min. To introduce trapped compounds into the gas chromatograph, the cold trap was then heated rapidly from −25 °C to 300 °C. The temperature of the valve and transfer line were maintained at 200 °C during analysis. Then, the whole system was baked at 300 °C for 3 min in preparation for the next sample analysis.

#### 2.3.3. Gas-Chromatography-Mass Spectrometry Analysis

Volatile organic compounds were identified using a 2010 GC coupled with an 8040 TQ-MS system (Shimadzu Corporation, Kyoto, Japan). The capillary column was a Shimadzu Rtx-5MS capillary column (30 m × 0.25 mm × 0.25 µm), and the carrier gas was helium at 1.0 mL/min. The split ratio was 1:40. The inlet temperature was 240 °C. The gradient temperature program was as follows: initial oven temperature was 40 °C, held for 3 min; 40–120 °C at 5 °C/min, held for 5 min; 120–240 °C at 30 °C/min, held for 8 min. The ionization mode of the MS was electron impact (EI). The temperatures of the interface and ion sources were 280 °C and 230 °C, respectively. The acquisition mode was full scan.

#### 2.3.4. Identification of Volatile Organic Compounds (VOC)

Shown in Table 1, the volatile compounds were identified by matching their mass spectra fragmentation patterns, retention index with those stored mass spectra libraries (NIST 11.L and Wiley 7), and combining them with existing works of literature [3,4,26,27,28]. The relative content of identified compounds was obtained by comparing them with the peak area of internal standards (Table 2).

### 2.4. Electronic nose (E-Nose) Measurements 

An ISENSO iNose E-nose system (Shanghai Ongshen Intelligent Technology Co., Ltd., Shanghai, China) was used to profile volatile fingerprints. Shown in Appendix A, the gas detectors of the E-nose system were composed of ten metal oxide sensors (MOS), each of which was sensitive to different volatile organic compounds [29], respectively.

A portion of each sample (3.0 g) was weighed into a headspace bottle (60 mL) and equilibrated in a 55 °C water bath for 40 min. Then, the gas in the headspace was pumped over the sensor surfaces for 5 min at a constant flow rate of 800 mL/min. Finally, cleaning the probe with continuously pumped filtered air until all sensors’ baseline value returned to 1.00 was preparation for the next sample analysis. The stable value of each sensor was extracted for data processing.

### 2.5. Statistical Analysis

Soft independent modelling by class analogy (SIMCA) 14.1 (Umetrics AB, Umea, Sweden) was used for partial least squares-discriminant analysis (PLS-DA) and principal component analysis (PCA). The heatmap and hierarchical cluster analysis (HCA) was conducted using MetaboAnalyst web (https://www.metaboanalyst.ca/MetaboAnalyst/faces/home.xhtml). Statistical package for the social sciences (SPSS) 21.0 (IBM, Chicago, IL, USA) were applied for multivariate statistical analysis. The differences among six grades of jasmine tea samples were estimated through analysis of variance (ANOVA). Regarding rank correlation analysis, the correlation between the response and grade of each substance was analyzed, the compounds with both a positive linear correlation or a negative correlation were screened out, respectively.

## 3. Results and Discussion

### 3.1. Identification of Volatile Organic Compounds in the Jasmine Tea by Automatic Thermal Desorption-Gas-Chromatography-Mass Spectrometry Spectrometry

To investigate the aroma characteristics of tested Jasmine tea samples, their volatile compounds were subjected to ATD-GC-MS, and the average relative amounts of identified volatiles were compared. Typical total ion chromatograms (TICs) are presented in Appendix A.

A total of 18 samples, with six different grades (named 1G, 2G, 2G, 3G, 4G, 5G, and 6G) and three repeats per category, were subjected to investigation of their aroma characteristics. The identified VOC and their corresponding amounts (mean ± standard deviation) were summarized; their significant differences also were tested (Table 2; Appendix A).

A total of sixty-three VOCs were identified (Table 1), including thirteen alcohols, five aldehydes, nineteen esters, twenty-three hydrocarbons, two ketones, one nitrogen compound, and one phenolic.

#### 3.1.1. Alcohols

There were thirteen kinds of identified alcohol in the jasmine tea grade samples. Among these identified alcohols, linalool and benzyl alcohol, which are abundant in jasmine flowers [4,27,30], accounted for 52.32% and 30.91% of the total content of alcohol, respectively.

Linalool, imparts a floral, fruity, and woody odor in jasmine tea, and benzyl alcohol provides a sweet, roasted, mild, fruity and citrus-like aroma, were contained in both the tea dhool and jasmine flowers [4,27,31]. Here, the relative content of 3-hexen-1-ol in alcohols was lower than both linalool and benzyl alcohol. Meanwhile, they were closely related to the sensory attributes of grassy and lettuce-like aromas [27,32]. Furthermore, among alcohols, there were some volatile compounds with a negative correlation to the grade of jasmine tea, including cyclopentanone, 1-hexanol, (Z)-Linalool oxide and (E)-Linalool oxide. These four volatile compounds are found in green tea, and existing studies show that cyclopentanone, (Z)-Linalool oxide and (E)-Linalool oxide are negatively correlated with the grade of green tea [16,32]. It also was reported that phenyl ethyl alcohol, α-Terpineol, and geraniol were all derived from jasmine flowers, having floral or sweet odor [3,4,33].

#### 3.1.2. Aldehydes

Five aldehydes, namely, benzaldehyde, decanal, hexanal, (E, E)-2,4-heptadienal and β-cyclocitral, were detected in all six grades of jasmine tea. Although aldehydes comprised 0.88% of the identified volatile organic compounds (VOC), they still contributed a lot to the aroma performance due to their low odor threshold [33]. Among aldehydes, benzaldehyde, which provided almond, sugar and burnt aroma notes, and decanal which supplied herbal, fatty and citrus aroma notes, were proven to play an essential role in aroma [4,26,27]. Here, all five aldehydes were negatively correlated with the grade of jasmine tea. Interestingly, these volatile compounds, which were harmful to the quality of jasmine tea, had been reported in green tea or originated from tea dhool [3,34,35]. Existing studies also demonstrated that hexanal and (E, E)-2,4-Heptadienal were negatively correlated with the grade of Japanese Matcha [34].

#### 3.1.3. Esters

Nineteen esters were found in all grades of jasmine tea. They accounted for 63.47% of identified volatile organic compounds (VOC) and positively correlated with the grade. Benzyl acetate, (Z)-3-hexanol benzoate, methyl salicylate, and their predecessors have confirmed methyl anthranilate as the main volatile aroma components of jasmine tea, which was consistent with the results of this study [3,4,9]. Among them were benzyl acetate, having floral, fruity odor notes, and (Z)-3-hexanol benzoate, with green, spicy, woody notes while herbaceous ones have prominent aroma characteristics of jasmine flowers [26,27,28]. Methyl anthranilate was described as similar to a peachy, sweet, fruity grape-like fragrance originated from jasmine flowers [3,4,27]. Methyl salicylate was considered to be a sweet, spicy, minty, wintergreen-like odor, and recognized as a significant aroma compound of black tea [3,36]. It is noteworthy that most volatile compounds of esters were positively correlated with the grade of jasmine tea and came from Jasmine flowers.

#### 3.1.4. Hydrocarbons 

Twenty-two hydrocarbons were identified in the tea samples. Despite the large number, it had a limited contribution to the aroma of tea [32,37]. Among them, α-farnesene, having floral and herbaceous odor notes, was the most abundant and recognized as one of the vital aroma components in jasmine tea [3,4,9]. Furthermore, Myrcene, Germacrene D and α-Farnesene, and so forth, were positively correlated with the grade of jasmine tea and were reported to originate from jasmine flowers [3,30], while α-pinene and limonene were negatively correlated with the grade.

#### 3.1.5. Ketones

Two ketones, namely 6-methyl-5-heptane-2-one and acetophenone, were identified. The 6-methyl-5-heptane-2-one was described as sweet, fruity, with orange odor notes, and previous studies confirmed that the compound showed an increasing trend in the processing of Oolong tea [9,38]. Regarding acetophenone, it was identified in Oolong tea [39], green tea [40], Pu’er tea [41] and Jasmine tea [27], but had little effect on the tea aroma. Moreover, there was a negative correlation between the relative content of 6-methyl-5-hapten-2-one and the quality grade.

#### 3.1.6. Nitrogen Compound

The nitrogen compound detected in the tea samples was indole, which provided nutty, floral, mothball, and burnt aroma notes. It was known as one of the main aroma components of jasmine tea and was positively correlated with the grade [3,4,8].

#### 3.1.7. Phenols

The phenol detected in the tea samples was eugenol. It may originate from the Jasmine Flower and be considered to be a clove-like spicy smell [8,9,26].

### 3.2. Multivariate Statistical Analysis of Identified Volatile Organic Compounds(VOC)

#### 3.2.1. Hierarchical Clustering Analysis (HCA)

To present VOC differences among different grade samples, a heat-map of eighteen samples versus identified compounds was plotted (Figure 1). The red color in the plot represents a higher content than the mean value; the blue color represents a lower content than the corresponding mean value. The HCA also was performed to get a cluster pattern among the six different grades. These six grades were subdivided into two categories, which were a high-grade group (including 1G, 2G, and 3G) and a low-grade group (including 4G, 5G, and 6G). By comparing the color intensity variation across all samples, we found that some compounds changed correlationally according to grade quality reduction (Appendix A).

Shown in Figure 1, the compounds marked with the blue frame, named **A**, indicated an increasing trend which correlated with the decline of grade. There was a total of twelve compounds, including five aldehydes (hexanal, decanal, β-Cyclocitral, benzaldehyde, (E,E)-2,4-Heptadienal), four alcohols (linalool oxide, 1-Hexanol, (Z)-Linalool oxide, cyclopentanol), two hydrocarbons (limonene, α-Pinene), and one ketone (5-Hepten-2-one), with fragrant characteristics such as fruity, floral, woody, green, sweet or more [4,26,27]. It was intriguing to find that most of these substances came from tea dhool [3,28]. Furthermore, according to existing research [16,34,35,42], most of them were negatively correlated to the quality of green tea.

Concerning the compounds in the red frame, **B,** (Figure 1), distinct rules were existing between the high-grade group (**1G, 2G,3G**) and the low-grade group (**4G,5G,6G**). Regarding the low-grade group, indicated as frame **B_1_**, positive linear correlations were existing. The amounts of both β-cadinene and (Z)-3-Hexenyl acetate, for example, were far higher in the higher grade. While, for the high-grade group, indicated as frame **B_2_**, there was not a simple linear relationship between their contents to the grade. Take β-cadinene as an example, the highest grade was in 1G, followed by 3G and 2G, however, Z-β-ocimene content was the highest in 2G, then in 1G and 3G. The reason may be that, in addition to the requirement of the intensity of flower fragrance, it is also an essential requirement for them to maintain Z-β-ocimene content at a moderate proportion, which could make its aroma coordinated.

Furthermore, as shown in Figure 1, there was a total of twenty-four compounds in frame **B**, including twelve esters, ten hydrocarbons, one alcohol, and one nitrogenous. It also is intriguing to find that most of them were absorbed from the jasmine flower [3,28,30]. It is remarkable that the main aroma components of jasmine tea (linalool, (Z)-3-Hexenyl benzoate, methyl salicylate, (Z)-3-Hexenyl acetate, α-Farnesene and indole) [3,4,9] were not linearly related to the grade, but were obviously rich in the high-grade group (1G, 2G,3G).

#### 3.2.2. Partial Least Square- Discriminant Analysis (PLS-DA)

A supervised PLS-DA was approached to investigate the differences among standard grade samples. Shown in Figure 2A, the scores of the principal component (PC) 1 (abscissa) and PC2 (ordinate) were new variables summarizing variables. The scores were orthogonal, which were completely independent of each other. The score of PC1 explains the largest variation of the X space, followed, by PC2. Hence, the scatter plot of PC1 versus PC2 was a window displaying how the X observations were situated concerning each other. Significant discrimination, according to the data matrix of the volatile compounds in the six grades, was observed. Two groups of tea samples with a higher grade (1G and 2G) were distributed in the fourth quadrant, three groups of tea samples with a lower grade (3G, 4G, 5G) were distributed in the first quadrant and the second quadrant, while the lowest group of tea samples (6G) were distributed in the third quadrant alone. The high grade explained the variance (R^2^Y = 0.966) and cross-validated predictive capability (Q^2^ = 0.979), manifesting the model’s feasibility.

Figure 2B reveals the result of cross-validation. The purpose of verification is to compare the goodness of fit (R^2^ and Q^2^) of the original model with that of multiple models based on data, where the order of y-observations is random and the x-matrix is complete. The low intercepts (R^2^ = 0.437, Q^2^ = –0.661) is an indication of the validity of the original model.

The PLS-DA loading scatter plot, Figure 2C, displays the relation between the X-variables and the Y-variables. Moreover, X-variables situated in the vicinity of the dummy Y-variables have the highest discriminatory power between the classes. Striking was that the plot in Figure 2C further explains the six grades of jasmine tea samples for differences in specific volatile components. Shown in Figure 2D, a total of thirty compounds were found with the VIP value over 1.0. The entire VIP values are ranked in Appendix A.

### 3.3. Response of Electronic Nose (E-Nose) Sensors to Volatile Organic Compounds (VOC) on Different Grades of Jasmine Tea 

The signals of ten sensors in response to VOC are presented in Figure 3. The Figure shows the signal response of S1 and S2 was far stronger than the rest of the sensors (S3–S10). Indicated by analysis of variance (ANOVA), except for S3, there were significant differences existing between the different grade samples. Looking at trends of correlation, it was found that the response signals of S1, S2, S6, and S10 were negatively correlated with the sample grade, while the signals of S4, S5, S7, S8, and S9 were positively correlated with the grade, which suggests that S1, S2, S6, and S10 could respond to a grade-negative VOC, while, S4, S5, S7, S8, and S9 could respond to a grade-positive VOC.

### 3.4. Multivariate Statistical Analysis of Electronic Nose (E-Nose) Sensor Response Signals 

#### 3.4.1. Hierarchical Clustering Analysis 

According to different correlation trends between the jasmine tea grades and the response signal intensity, all ten sensors can be subdivided into three categories, which were a negative correlation, positive correlation, and irrelevance.

Shown in the Frame A (Figure 4), for S1 (sensitive to Ammonia and Amines), S2 (Hydrogen sulfide and sulfides), S6 (Methane, ethane and hydrocarbons), and S10 (Alkanes and flammable gases), there was an apparent negative correlation between their signal intensity to the jasmine tea grade. The higher the response value was, in other words, the lower it was in its grade. The types of volatile organic compounds (VOC) they were sensitive to coincided with components negatively related to the jasmine tea quality.

The second type, as indicated in Frame B (Figure 4), includes S4 (Alcohol and Organic Solvents), S5 (Volatile gases in food cooking), S7 (Flammable gases) and S8 (Volatile Organic Compounds) as their signal response intensity was positively correlated with the tea grading, which meant they reflected the content of volatiles positively related to the jasmine tea grade, so we could name them as positive VOC recognition sensors.

The rest of the sensors, S3 (hydrogen) and S9 (Hydroxide, gasoline, and kerosene), were irrelevant sensors for evaluating jasmine tea aroma, as there was no regularity in the signals appearing in response to the grade. This result was reasonable because the corresponding sensitive gas does not exist in jasmine tea at all. Therefore, both S3 and S9 should be ignored to reduce data noise.

#### 3.4.2. Principal Component Analysis (PCA)

After removing signals from both the S3 and S9 sensors, the electronic nose (E-nose) data was subjected to PCA analysis, through which we could obtain an overview of sample similarity. Shown in Figure 5, PC1 and PC2 explain 59.9% and 33.1% of the total variance, respectively. It was intriguing to find that the 1G, 2G, 3G, and 4G samples were not distinguished completely, whereas there was a clear separation of 5G and 6G samples from the other grade samples.

After comparing the difference in sensory evaluation criteria of these grade samples, we found that it was reasonable. Rather than a significant difference in the aroma intensity [6] (Appendix A), the main difference for samples in area **I** (1G, 2G, 3G, and 4G) were certain specific characteristics, such as the freshness and durability of the aroma. Therefore, it indicates that areas **I** (1G, 2G, 3G, and 4G), **II** (5G), and **III** (6G) were mainly reflecting aroma concentration. It also suggests that the E-nose could be good at recognizing aroma concentration but may not good at identifying specific unique aroma characteristics of high-grade jasmine tea. The following two reasons may attribute to this conclusion. First, the strength of volatile components that have a pivotal contribution to freshness and persistence was deficient and could not respond well to these sensors. Second, the formation of freshness and durability were not determined by some specific volatile substances, but by the combination of some elements within a particular range of proportion.

## 4. Conclusions

A group of authoritative jasmine tea grade samples, which were prepared following Chinese National Standard requirements, were subjected to research. Both Automatic thermal desorption-gas-chromatography-mass spectrometry (ATD-GC-MS) and electronic nose (E-nose) were applied for discrimination and were compared systematically.

Consequently, a total of sixty-three volatile compounds were tentatively identified by ATD-GC-MS. Through both partial least square-discriminant analysis (PLS-DA) and hierarchical cluster analysis (HCA), a satisfactory discriminant result was achieved. Twelve of these compounds, including four alcohols, five aldehydes, two hydrocarbons, and one ketone, were found to be negatively correlated to the jasmine tea grade. It is worth noting that most of the main aroma components of jasmine tea, such as linalool, (Z)-3-Hexenyl benzoate, methyl salicylate, (Z)-3-Hexenyl acetate, α-Farnesene and indole, have no linear relationship between their contents to the tea grade, but are obviously abundant in the high grade.

Regarding the electronic nose, the signal intensities of S1 (sensitive to Ammonia and Amines), S2 (Hydrogen sulfide and sulfides), S6 (Methane, ethane and hydrocarbons), and S10 (Alkanes and flammable gases) were negatively correlated to the tea grades. While, S4 (Alcohol and Organic Solvents), S5 (Volatile gases in food cooking), S7 (Flammable gases) and S8 (Volatile Organic Compounds), were positively correlated to the tea grades. It was interesting to find that the E-nose was better at detecting aroma concentrations rather than recognizing unique aroma characteristics.

## Figures and Tables

**Figure 1 molecules-25-00380-f001:**
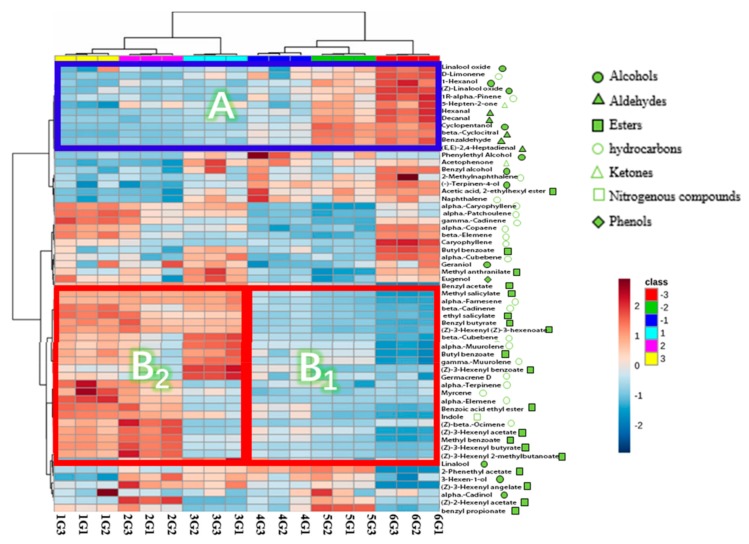
Heatmap of volatile compounds in six grades of jasmine tea. (Note: 1G1, 1G2, and 1G3 represent three repeats of the first-grade jasmine tea; so to the followings grade samples).

**Figure 2 molecules-25-00380-f002:**
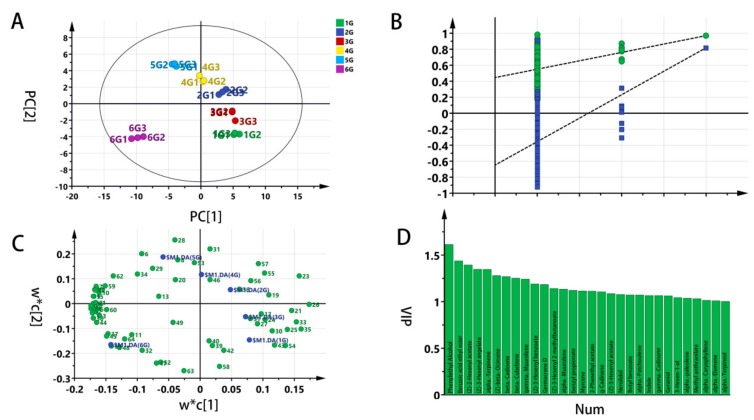
Partial least square-discriminant analysis (PLS-DA) of jasmine tea samples with soft independent modelling by class analogy (SIMCA). (**A**) PLS-DA scores scatter plot with pareto scaling mode (R^2^Y = 0.966 and Q^2^ = 0.979); (**B**) The result of the cross-validation model with 200 times of calculations using a permutation test (R^2^ = 0.437, Q^2^ = –0.661); (**C**) PLS-DA loading scatter plot (R^2^X[1] = 0.498 R^2^X[2] = 0.181); (**D**) The variable importance for projection (VIP) plot (VIP >1).

**Figure 3 molecules-25-00380-f003:**
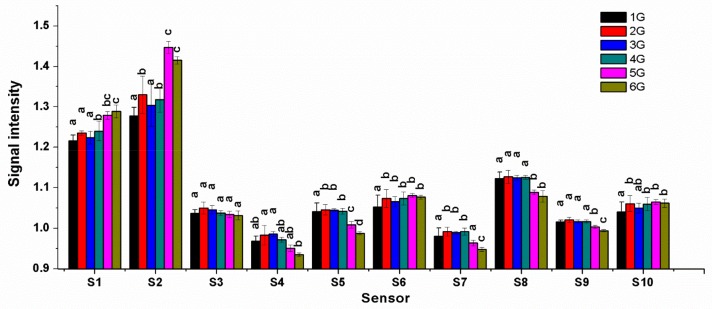
Response values of ten sensors to volatile compounds from different grades of jasmine tea samples. *Note: The bar marked with the same letter (**a**,**b**,**c**), within a sensor, are not significantly different between two grade samples (*p* > 0.05).

**Figure 4 molecules-25-00380-f004:**
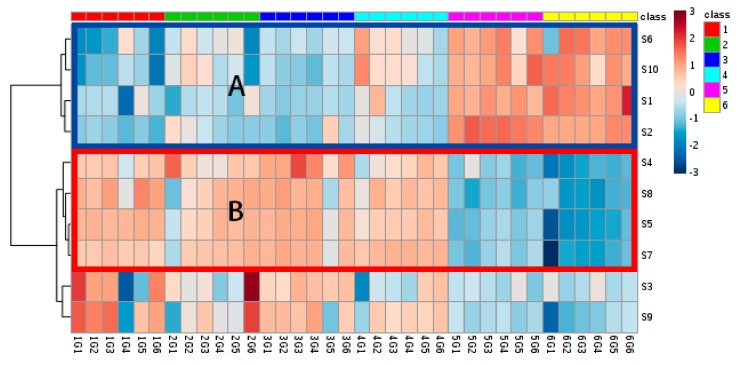
Heatmap of stable signals of E-nose sensors for six grades of jasmine tea (*Note: S1–S10 represent the ten sensors of the E-nose; 1G1, 1G2, 1G3, 1G4, 1G5, and 1G6 represent the six repeats of grade 1 jasmine tea, as do the following grade samples; The blue frame (**A**) indicates sensors with a negative correlation to the grade; The red frame (**B**) indicates sensors with a positive correlation to the grade).

**Figure 5 molecules-25-00380-f005:**
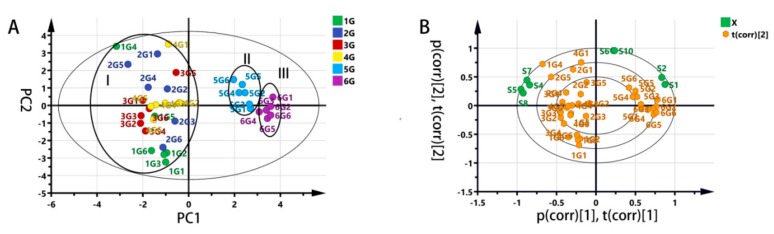
Principal component analysis (PCA) of jasmine tea samples. (**A**) PCA scores scatter plot with Pareto scaling mode (R2X [1] = 0.599 and R2X [2] = 0.331); (**B**) Biplot of ten sensors and standard jasmine tea samples with Pareto scaling mode.

**Table 1 molecules-25-00380-t001:** The identification of Volatile Organic Compounds (VOC) in grade jasmine tea.

No.	Compound	CAS ^g^	RT ^g^	RI ^g^	MS Fragments			MS ^g^
1	Cyclopentanol	96-41-3	5.004	788	57	44	41	83
2	1-Hexanol	111-27-3	7.893	860	56	43	41	85
3	Benzyl alcohol	100-51-6	13.430	1036	79	108	107	95
4	(Z)-Linalool oxide	5989-33-3	14.665	1072	59	94	43	93
5	(E)-Linalool oxide	34,995-77-2	15.205	1088	59	94	43	97
6	3-Hexen-1-ol	928-96-1	7.432	868	67	41	39	95
7	Linalool	78-70-6	15.775	1104	71	93	55	93
8	Phenylethyl Alcohol	60-12-8	16.050	1117	91	92	65	95
9	(-)-Terpinen-4-ol	20,126-76-5	18.265	1137	71	111	43	87
10	α-Terpineol	98-55-5	18.691	1143	59	93	121	88
11	Geraniol	106-24-1	20.455	1228	69	41	48	90
12	Nerolidol	7212-44-4	27.715	1564	41	69	43	84
13	α-Cadinol	481-34-5	28.526	1580	95	121	43	90
14	Hexanal	66-25-1	5.780	806	44	56	41	96
15	Benzaldehyde	100-52-7	10.915	982	77	106	105	96
16	(E,E)-2,4-Heptadienal	4313-03-5	12.621	1011	81	110	41	90
17	Decanal	112-31-2	18.980	1204	43	41	57	96
18	β-Cyclocitral	432-25-7	19.437	1218	137	152	109	93
19	(Z)-3-Hexenyl acetate	3681-7-18	12.425	1002	43	67	82	95
20	(Z)-2-Hexenyl acetate	56,922-75-9	12.757	1005	43	67	82	87
21	Methyl benzoate	93-58-3	15.405	1060	105	77	136	98
22	Acetic acid, 2-ethylhexyl ester	103-09-3	17.130	1149	43	70	57	84
23	Benzyl acetate	140-11-4	17.750	1162	108	91	90	93
24	Benzoic acid ethyl ester	93-89-0	17.915	1171	105	77	122	88
25	(Z)-3-Hexenyl butyrate	16,491-36-4	18.342	1182	82	67	71	80
26	Methyl salicylate	119-36-8	18.635	1191	120	92	152	96
27	(Z)-3-Hexenyl 2-methylbutanoate	53,398-85-9	19.754	1226	67	82	57	80
28	2-Phenethyl acetate	103-45-7	20.570	1249	104	43	91	93
29	benzyl propionate	122-63-4	20.675	1259	91	108	57	75
30	ethyl salicylate	118-61-6	21.161	1270	120	92	166	91
31	(Z)-3-Hexenyl angelate	84,060-80-0	23.486	1282	82	55	67	91
32	Methyl anthranilate	134-20-3	24.346	1343	119	92	151	96
33	Benzyl butyrate	103-37-7	24.497	1346	108	91	178	83
34	Butyl benzoate	136-60-7	25.261	1359	105	123	77	82
35	(Z)-3-Hexenyl (Z)-3- hexenoate	61,444-38-0	25.455	1388	82	67	69	92
36	(Z)-3-Hexenyl benzoate	25,152-85-6	27.825	1565	105	67	77	97
37	Benzyl Benzoate	120-51-4	29.302	1733	105	91	77	76
38	α-Pinene	7785-70-8	10.005	948	93	92	91	85
39	Myrcene	123-35-3	11.921	958	41	93	69	86
40	α-Terpinene	99-86-5	12.846	1016	121	93	136	90
41	Limonene	5989-27-5	13.261	1018	68	93	67	88
42	(Z)-β-Ocimene	13,877-91-3	13.845	1031	93	91	79	93
43	α-Elemene	20,307-84-0	24.097	1340	121	93	136	85
44	α-Cubebene	17,699-14-8	24.645	1351	161	105	119	93
45	α-Copaene	3856-25-5	25.365	1373	161	119	105	90
46	Germacrene D	23,986-74-5	25.530	1477	161	105	91	89
47	γ-Cadinene	39,029-41-9	25.600	1514	161	204	105	90
48	β-Elemene	515-13-9	25.626	1398	81	93	68	80
49	α-Gurjunene	489-40-7	25.960	1413	204	161	105	87
50	Caryophyllene	87-44-5	26.155	1494	93	133	91	82
51	β-Cubebene	13,744-15-5	26.306	1387	161	105	91	92
52	α-Caryophyllene	6753-98-6	26.660	1579	93	80	41	88
53	γ-Muurolene	3002-74-0	26.885	1435	161	105	119	88
54	α-Farnesene	502-6-4	27.205	1458	41	93	69	93
55	β-Cadinene	483-76-1	27.365	1469	161	134	119	90
56	α-Muurolene	10,208-80-7	27.545	1479	105	161	94	89
57	α-Patchoulene	560-32-7	27.976	1460	135	93	107	78
58	Naphthalene	91-20-3	18.381	1231	128	129	127	97
59	2-Methylnaphthalene	91-57-6	22.350	1345	142	141	115	86
60	5-Hepten-2-one	110-93-0	11.735	938	43	41	69	93
61	Acetophenone	98-86-2	14.436	1068	105	77	51	97
62	Indole	120-72-9	22.107	1340	117	90	89	97
63	Eugenol	97-53-0	24.712	1392	164	103	77	88

^g^ CAS: Chemical Abstracts Service; RT: Retention time; RI: Retention index; MS: Match score of mass spectra libraries.

**Table 2 molecules-25-00380-t002:** VOC of grade jasmine tea.

No.	MI^f^	Compound	Average Relative Content (×10 μg/g)					
			1G^e^	2G^e^	3G^e^	4G^e^	5G^e^	6G^e^
1	MS,RI	Cyclopentanol	0.04 ± 0.00b	0.25 ± 0.02b	0.13 ± 0.03b	0.59 ± 0.15ab	0.89 ± 0.25a	1.02 ± 0.22a
2	MS,RI	1-Hexanol	0.04 ± 0.00b	0.06 ± 0.01b	0.11 ± 0.02b	0.10 ± 0.03b	0.09 ± 0.02b	0.23 ± 0.08a
3	MS,RI	Benzyl alcohol	5.38 ± 0.73b	37.76 ± 19.39a	6.69 ± 1.09b	12.27 ± 4.05b	4.71 ± 1.73b	9.14 ± 1.41b
4	MS,RI	(Z)-Linalool oxide	0.22 ± 0.04a	0.32 ± 0.01a	0.27 ± 0.05a	0.36 ± 0.09a	0.30 ± 0.11a	0.50 ± 0.09a
5	MS,S,RI	(E)-Linalool oxide	0.71 ± 0.12a	1.09 ± 0.05a	0.89 ± 0.16a	0.84 ± 0.20a	0.61 ± 0.23a	0.73 ± 0.12a
6	MS,RI	3-Hexen-1-ol	2.82 ± 0.26b	5.95 ± 0.72ab	6.32 ± 1.43a	5.15 ± 1.30ab	2.92 ± 0.88b	2.23 ± 0.52b
7	MS,S,RI	Linalool	15.50 ± 1.04b	37.60 ± 1.03a	26.65 ± 5.61ab	24.19 ± 6.96ab	10.76 ± 3.19b	13.84 ± 2.73b
8	MS,RI	Phenylethyl Alcohol	0.04 ± 0.02b	0.80 ± 0.36a	0.02 ± 0.01b	0.79 ± 0.28a	0.18 ± 0.12ab	0.08 ± 0.02b
9	MS,RI	(-)-Terpinen-4-ol	0.01 ± 0.00b	0.05 ± 0.02a	0.01 ± 0.00b	0.03 ± 0.01ab	0.02 ± 0.01b	0.02 ± 0.00ab
10	MS,RI	α-Terpineol	0.12 ± 0.00ab	0.21 ± 0.01a	0.11 ± 0.02ab	0.21 ± 0.06a	0.09 ± 0.03b	0.11 ± 0.01ab
11	MS,RI	Geraniol	0.27 ± 0.05ab	0.45 ± 0.03a	0.41 ± 0.13ab	0.19 ± 0.06b	0.15 ± 0.06b	0.21 ± 0.05ab
12	MS,RI	Nerolidol	0.14 ± 0.00b	0.30 ± 0.01a	0.12 ± 0.02b	0.11 ± 0.03b	0.03 ± 0.01c	0.04 ± 0.01c
13	MS,RI	α-Cadinol	0.01 ± 0.00a	0.04 ± 0.02a	0.01 ± 0.00a	0.01 ± 0.00a	0.00 ± 0.00a	0.00 ± 0.00a
14	MS,RI	Hexanal	0.05 ± 0.00b	0.21 ± 0.03ab	0.09 ± 0.02b	0.20 ± 0.07ab	0.20 ± 0.05ab	0.36 ± 0.07a
15	MS,RI	Benzaldehyde	0.25 ± 0.02b	0.59 ± 0.11ab	0.36 ± 0.08ab	0.58 ± 0.16ab	0.56 ± 0.16ab	0.74 ± 0.15a
16	MS,RI	(E,E)-2,4-Heptadienal	0.05 ± 0.00b	0.09 ± 0.00b	0.09 ± 0.02b	0.20 ± 0.05b	0.40 ± 0.12ab	0.60 ± 0.12a
17	MS,RI	Decanal	0.17 ± 0.01b	0.34 ± 0.09b	0.21 ± 0.07b	0.34 ± 0.09b	0.43 ± 0.10ab	0.70 ± 0.14a
18	MS,RI	β-Cyclocitral	0.08 ± 0.00b	0.20 ± 0.02ab	0.13 ± 0.03ab	0.24 ± 0.07ab	0.23 ± 0.07ab	0.26 ± 0.05a
19	MS,RI	(Z)-3-Hexenyl acetate	3.07 ± 0.30a	1.47 ± 0.16ab	4.93 ± 1.10a	2.44 ± 0.63ab	1.07 ± 0.30b	0.23 ± 0.05b
20	MS,RI	(Z)-2-Hexenyl acetate	0.03 ± 0.00b	0.02 ± 0.00b	0.09 ± 0.02a	0.04 ± 0.01b	0.03 ± 0.01b	0.02 ± 0.00b
21	MS,RI	Methyl benzoate	11.17 ± 0.90ab	8.46 ± 0.57b	16.96 ± 3.46a	8.02 ± 2.14b	2.65 ± 0.79bc	0.92 ± 0.22c
22	MS,RI	Acetic acid, 2-ethylhexyl ester	0.03 ± 0.00b	0.08 ± 0.02a	0.03 ± 0.00b	0.07 ± 0.02ab	0.03 ± 0.01b	0.04 ± 0.01ab
23	MS,RI	Benzyl acetate	35.70 ± 2.87a	57.80 ± 2.39a	52.12 ± 10.66a	37.08 ± 10.04a	12.84 ± 3.90b	4.24 ± 1.45b
24	MS,RI	Benzoic acid ethyl ester	0.02 ± 0.00ab	0.03 ± 0.00a	0.02 ± 0.00ab	0.01 ± 0.00ab	0.01 ± 0.00b	0.01 ± 0.00b
25	MS,RI	(Z)-3-Hexenyl butyrate	0.41 ± 0.03b	0.18 ± 0.05c	0.63 ± 0.13a	0.15 ± 0.04c	0.06 ± 0.02c	0.02 ± 0.00c
26	MS,S,RI	Methyl salicylate	10.45 ± 0.72ab	15.45 ± 1.13a	13.25 ± 2.45a	7.02 ± 1.86b	2.58 ± 0.82bc	1.11 ± 0.23c
27	MS,RI	(Z)-3-Hexenyl 2-methylbutanoate	0.18 ± 0.01b	0.13 ± 0.01bc	0.30 ± 0.06a	0.13 ± 0.03bc	0.06 ± 0.02c	0.05 ± 0.01c
28	MS,RI	2-Phenethyl acetate	0.29 ± 0.02ab	0.60 ± 0.01a	0.43 ± 0.07ab	0.60 ± 0.16a	0.35 ± 0.12ab	0.12 ± 0.02b
29	MS,RI	Benzyl propionate	0.02 ± 0.00ab	0.03 ± 0.00ab	0.03 ± 0.01a	0.03 ± 0.01ab	0.02 ± 0.01ab	0.01 ± 0.00b
30	MS,RI	Ethyl salicylate	0.08 ± 0.00a	0.09 ± 0.00a	0.08 ± 0.01a	0.04 ± 0.01b	0.02 ± 0.01b	0.02 ± 0.00b
31	MS,RI	(Z)-3-Hexenyl angelate	0.21 ± 0.01b	0.41 ± 0.02a	0.39 ± 0.07a	0.27 ± 0.07ab	0.16 ± 0.05b	0.08 ± 0.01b
32	MS,RI	Methyl anthranilate	4.84 ± 0.25b	15.64 ± 3.15a	6.38 ± 0.94b	4.76 ± 1.35b	1.39 ± 0.55b	3.47 ± 1.20b
33	MS,RI	Benzyl butyrate	0.06 ± 0.01a	0.08 ± 0.00a	0.08 ± 0.01a	0.03 ± 0.01b	0.02 ± 0.01b	0.01 ± 0.00b
34	MS,RI	Butyl benzoate	0.08 ± 0.00b	0.13 ± 0.00a	0.09 ± 0.02ab	0.09 ± 0.02ab	0.05 ± 0.02b	0.04 ± 0.01b
35	MS,RI	(Z)-3-Hexenyl (Z)-3-hexenoate	0.20 ± 0.01a	0.26 ± 0.00a	0.25 ± 0.04a	0.09 ± 0.02b	0.03 ± 0.01b	0.02 ± 0.01b
36	MS,RI	(Z)-3-Hexenyl benzoate	7.09 ± 0.19bc	23.58 ± 0.95a	8.62 ± 1.22b	9.25 ± 2.63b	2.51 ± 0.87c	2.47 ± 0.61c
37	MS,RI	Benzyl Benzoate	0.02 ± 0.00b	0.05 ± 0.01a	0.02 ± 0.00b	0.03 ± 0.01b	0.01 ± 0.00b	0.03 ± 0.01ab
38	MS,RI	α-Pinene	0.05 ± 0.01a	0.07 ± 0.01a	0.07 ± 0.01a	0.08 ± 0.03a	0.06 ± 0.01a	0.11 ± 0.02a
39	MS,RI	Myrcene	0.10 ± 0.02a	0.07 ± 0.01ab	0.09 ± 0.02ab	0.05 ± 0.01b	0.03 ± 0.01b	0.03 ± 0.01b
40	MS,RI	α-Terpinene	0.01 ± 0.00a	0.01 ± 0.00ab	0.01 ± 0.00a	0.01 ± 0.00ab	0.00 ± 0.00b	0.00 ± 0.00b
41	MS,RI	Limonene	0.11 ± 0.01b	0.33 ± 0.10a	0.14 ± 0.03b	0.17 ± 0.05ab	0.11 ± 0.04b	0.15 ± 0.03ab
42	MS,RI	(Z)-β-Ocimene	0.14 ± 0.01b	0.06 ± 0.01c	0.27 ± 0.05a	0.06 ± 0.01c	0.03 ± 0.01c	0.06 ± 0.01c
43	MS,RI	α-Elemene	0.04 ± 0.00a	0.01 ± 0.00c	0.03 ± 0.01b	N.D.^f^	N.D.^f^	N.D.^f^
44	MS,RI	α-Cubebene	0.12 ± 0.01b	0.20 ± 0.01a	0.13 ± 0.03ab	0.12 ± 0.03ab	0.07 ± 0.02b	0.12 ± 0.02b
45	MS,RI	α-Copaene	0.32 ± 0.02ab	0.40 ± 0.01a	0.33 ± 0.06a	0.27 ± 0.07ab	0.15 ± 0.05b	0.26 ± 0.05ab
46	MS,RI	Germacrene D	0.01 ± 0.00b	0.01 ± 0.00a	0.01 ± 0.00b	0.01 ± 0.00b	0.00 ± 0.00b	0.00 ± 0.00b
47	MS,RI	γ-Cadinene	0.18 ± 0.01a	0.15 ± 0.01ab	0.18 ± 0.03a	0.11 ± 0.03ab	0.05 ± 0.02b	0.10 ± 0.02b
48	MS,RI	β-Elemene	0.10 ± 0.01a	0.10 ± 0.00a	0.10 ± 0.02a	0.08 ± 0.02ab	0.04 ± 0.02b	0.08 ± 0.01ab
49	MS,RI	α-Gurjunene	0.01 ± 0.00a	0.02 ± 0.01a	0.01 ± 0.00a	0.00 ± 0.00a	0.00 ± 0.00a	0.01 ± 0.00a
50	MS,RI	Caryophyllene	0.13 ± 0.01a	0.11 ± 0.00ab	0.12 ± 0.02ab	0.11 ± 0.03ab	0.05 ± 0.02b	0.18 ± 0.03a
51	MS,RI	β-Cubebene	0.31 ± 0.02ab	0.41 ± 0.01a	0.30 ± 0.05ab	0.23 ± 0.06b	0.10 ± 0.03b	0.10 ± 0.02b
52	MS,RI	α-Caryophyllene	0.37 ± 0.02ab	0.51 ± 0.02a	0.39 ± 0.07ab	0.26 ± 0.07b	0.12 ± 0.04b	0.21 ± 0.03b
53	MS,RI	γ-Muurolene	0.23 ± 0.02b	0.41 ± 0.01a	0.24 ± 0.05b	0.22 ± 0.06b	0.13 ± 0.04bc	0.08 ± 0.01c
54	MS,RI	α-Farnesene	6.56 ± 0.42bc	12.54 ± 1.29a	8.80 ± 1.59b	4.10 ± 1.14c	1.02 ± 0.14c	1.62 ± 0.45c
55	MS,RI	β-Cadinene	1.22 ± 0.06b	2.39 ± 0.02a	1.21 ± 0.20b	1.02 ± 0.29bc	0.51 ± 0.18c	0.35 ± 0.04c
56	MS,RI	α-Muurolene	0.16 ± 0.01b	0.33 ± 0.00a	0.16 ± 0.03b	0.15 ± 0.04bc	0.08 ± 0.03c	0.03 ± 0.00c
57	MS,RI	α-Patchoulene	0.05 ± 0.00a	0.06 ± 0.00a	0.05 ± 0.01a	0.02 ± 0.01b	0.01 ± 0.00b	0.02 ± 0.00b
58	MS,RI	Naphthalene	0.41 ± 0.02b	1.41 ± 0.29a	0.48 ± 0.07b	1.15 ± 0.36ab	0.44 ± 0.15b	0.51 ± 0.04b
69	MS,RI	2-Methylnaphthalene	0.06 ± 0.00b	0.15 ± 0.05a	0.08 ± 0.01ab	0.12 ± 0.04ab	0.05 ± 0.02b	0.07 ± 0.00ab
60	MS,RI	6-Methyl-5-hepten-2-one	0.71 ± 0.05b	1.61 ± 0.12ab	1.62 ± 0.35a	1.12 ± 0.30ab	0.93 ± 0.25ab	1.38 ± 0.31ab
61	MS,RI	Acetophenone	0.31 ± 0.01b	1.48 ± 0.24a	0.33 ± 0.05b	1.03 ± 0.33ab	0.40 ± 0.14b	0.42 ± 0.04b
62	MS,RI	Indole	7.20 ± 0.27ab	14.19 ± 3.36ab	1.78 ± 0.82b	20.55 ± 11.49a	1.45 ± 0.07b	0.70 ± 0.24b
63	MS,RI	Eugenol	0.12 ± 0.00b	0.25 ± 0.04a	0.11 ± 0.02b	0.10 ± 0.03b	0.05 ± 0.02b	0.06 ± 0.02b

a–d Means ± SD followed by the same letter, within a row, are not significantly different (*p* > 0.05); e 1G, 2G, 3G, 4G, 5G, and 6G represent the standard sample for the grade of jasmine tea from high rank to low rank; f MI, method of identification; N.D., peak intensity lower than triple signal-to-noise.

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
