# Peer review of "Comparison of Volatiles in Different Jasmine Tea Grade Samples Using Electronic Nose and Automatic Thermal Desorption-Gas Chromatography-Mass Spectrometry Followed by Multivariate Statistical Analysis"

_molecules, 2020, doi:10.3390/molecules25020380_

Round 1
Reviewer 1 Report
Sensory assessment is usually used to evaluate quality or grading of jasmine scented teas. The authors tried to discriminate the grading of jasmine scented tea samples using GC/MS and E-nose. It will be helpful to develop instrumental evaluation of jasmine tea samples. However, the manuscript has not been ready for publication in the present form. I recommend its publication after major revision.
There have been many studies on the instrumental assessments of jasmine scented tea quality, the related references are not updated in this manuscript. Such as:Liang YR et al (2007). Application of chemical composition and infusion colour difference analysis to quality estimation of jasmine-scented tea. DOI: 10.1111/j.1365-2621.2007.01267.x.
Lin J et al (2013). A novel quality evaluation index and strategies to identify scenting quality of jasmine tea based on headspace volatiles analysis. DOI: 10.1007/s10068-013-0085-x.
Shen JX et al (2017). Differential Contribution of Jasmine Floral Volatiles to the Aroma of Scented Green Tea. DOI: 10.1155/2017/5849501.
The significance of the study was not clarified in the INTRODUCTION. The experimental design was not sound. The volatile compounds of the jasmine flower vary with production seasons and production regions, and the volatile composition of the jasmine scented teas depend on the source of flowers. The tested samples were from one source (Fujian Tea Import and Export Co.) and they could not represent those from other sources, such as Hengxian County, Guangxi Province. The conclusion is weak, such as: “Partial least squares-discriminant analysis (PLS-DA) and hierarchical cluster analysis (HCA) showed a satisfactory discriminant effect on grade”. “The amounts of such components could affect the coordination of aroma, which would result in difference of aroma grade for high-level group”. “ E-nose was good at recognizing aroma concentration of jasmine tea, but not...”. However, the readers did not get know how to discriminate the grades of the tea samples based on the manuscript information. The contribution of the study to the knowledge of related research field has not been clarified. The authors should compare their own works with the published works to show their contribution to the related areas.Author Response
Please see the attachment

Reviewer 2 Report
This study shows some interesting approach and results on the comparison of volatiles in jasmine tea according to grade.
However, some points showed be added and revised before a publication.
Line 108, more information on the internal standard compounds such as amount and concentration added. Line 104-108, The identification of most volatiles was made based on only on-computer MS libraries. Other ID methods such as positive identification using authentic standards and retention index (RI) values should be included for the identification Regarding quantification, I assume all the volatiles studied have different recoveries and detector response. Accordingly, the absolute concentrations obtained in this study are not correct without proper calibrations. I also like to ask authors to provide any validation results on their analytical procedure. How were those positive and negative correlators selected? Please, add more explanation on it. Line 185, Both isomers could not be separated on 5-type columns. Alpha-pine and limonene would be correct. In figure 2, only 6G samples would be separated from others. In figure 5, area II and III would not be clearly discriminated in my opinion.
Reviewer 3 Report
In this paper, the authors compare VOC pattern in different grade samples of Chinese jasmine tea through automatic thermal desorption-gas chromatography-mass spectrometry (ATD-GC-MS) and E-nose. Multivariate analysis allowed to identify several VOCs negatively or positively correlated to jasmine tea grade and highlights that E-nose was good to recognize aroma concentration.
The work is well conducted but requires some corrections before publication. Some of these are very important and without these the paper cannot be published.
In the title, “Depending Volatiles” is not immediately understandable. I suggest a new title. For example, “Comparison of Volatiles in different Jasmine tea grade samples….” The sentence “Chinese jasmine tea is a type of flower-scented tea, which produced by mixing green tea 15 with Jasminum sambac flower repeatedly” is repeat in abstract and in introduction. I suggest you to it in one of the two point. When used the term “VOC” us adjective, the singular must be used. The authors must correct these errors (i.e. VOC difference, VOC concentration, VOC extraction etc.) in all paper. Abstract:Line 18. Add “Organic” to Volatile compound because VOC abbreviation include it.
Line 19 and 50. Change “gas-chromatogram” with “gas-chromatography”.
Line 38. Dhool, not adhool.
Key words: Add “Organic” to “volatile compounds” Materials and methods:Line 66. Replace grades with grade.
Section 2.4. The authors must explain:
What sensors read? Resistance or current? How the sensor answer was calculated? Results and discussionLine 136. There are two “corresponding”.
Table 1. Amount of single VOC is relative to 10 µg/g of volatiles captured by thermal desorption. What is the unit of measure for compounds?
It is very important to report for each compound: CAS number, retention time, NIST match, experimental m/z. Without these parameters, the publication of this paper cannot happen.
In same case, ND is reported. What is limit of detection?
The use of letters (a,b,c) to indicate significantly different is not very clear. Could the authors change this with something more immediate understanding?
Line 157. Replace (E,E)-2,4-heptagonal with (E,E)-2,4-heptadienal.
Line 158. Replace cycloconcitral with cyclocitral.
Line 203. Replace verses with versus.
Line 204. Colour scale which is used in this analysis is red to blue, not green!
Figure 1. For 3G grade is reported twice 3G3.
The meaning of arrows is not explained. I suggest to eliminate they. Colour are sufficient to understand.
Line 221. High-level group (1G,2G,3G) are indicated as frame B2, not B1.
Lines 222-224. The sentence is not clear. Change it, please!
Line 239. PC is principal component, not primary!
Figure 2. Put A,B,C,D letters out of graphics.
In the text, the authors call PC1 and PC2, while in figure t(1) e t(2). Uniform the names.
Figure 4. I’m very confused. All study design is based on three replicates. In this figure 6 replicates are reported. Eliminate also in this figure the allows.
Figure 5. The circles that identify area I,II and III are shift and not locate the right zone.
Reference:Reference 5 is not well formatted.
Round 2
Reviewer 1 Report
Many concerns were addressed.
However, the conclusions are also very weak. Many conclusion sentences are common sense, not the conclusions drawn from the research results, for example: ATD-GC-MS could provide profound information on the VOC; ... Although the information provided by the electronic nose seemed limited, it was still a convenient method in rank distinguishing; ... Appling different technicals to distinguish the same grade samples and making a comprehensive comparison will help us to understand...
Standard English should be used. The readers can't understand what the authors meant: Standard Chinese jasmine tea grade samples; high-level tea; low-level tea; ...
Author Response
Reviewer 1
1. The conclusions are also very weak. Many conclusion sentences are common sense, not the conclusions drawn from the research results, for example: ATD-GC-MS could provide profound information on the VOC; ... Although the information provided by the electronic nose seemed limited, it was still a convenient method in rank distinguishing; ... Appling different technicals to distinguish the same grade samples and making a comprehensive comparison will help us to understand
Thank you! We have rewritten the conclusion part accordingly (Line 342-357).
Standard English should be used. The readers can't understand what the authors meant: Standard Chinese jasmine tea grade samples; high-level tea; low-level tea;
Thank you! Correct accordingly.
“Standard Chinese jasmine tea grade samples” has been modified to “A group of authoritative jasmine tea grade samples” (Line 77, 342)
“high-level tea” has been modified to “high-grade tea” (Line 222, 237, 240, 251,336)
“low-level tea” has been modified to “low-grade tea” (Line 223, 238)
Reviewer 2 Report
Some responses and revisions are generally acceptable.
However, it is still regrettable that the identification of all the volatile compounds was made without using authentic standard compounds, although most compounds detected in this study would be commercially available.
The positive identification of volatiles (at least) related to the discrimination of jasmine teas could reinforce the results of this study.
Regarding the discrimination of samples of different grades,
I cannot see any consistent grouping and discrimination of samples from 1st to 5th grades in figure 2.
Also, in figure 5, samples cannot be discriminated according to their grades in group I.
Other minor comments are as follows.
Line 73, compose? Is that ‘composition’? The nomenclatures of chemicals would not need to start with capital letters For example, ‘Methyl salicylate’, ‘Linalool’, ‘Linalool oxide, line 83-84 ; line 95, ‘weight’should be replaced by ‘weighted’. Line 104-105, line 106, line 128-129 : Please, rewrite these sentences. Line 119, ‘Retention index’ needs no ‘capital letter’. Table 1, No. 4 and No 5. have the same RI values although they showed the different retention time. Please, check them.Also, the Greek alphabets should be used for ‘alpha’, ‘beta’ and so on throughout the manuscript including tables. What do you mean the Experimental m/z? Please, confirm the nomenclature of ‘g-Cadinene’. How did you obtain the relative contents at the concentrations of x10 ug/g? Line 222, ‘correlational’ should be replaced by ‘correlationally’. table S2, 62 would be (E)-Linalool oxide. Table S1 and S2 need references.Author Response
Reviewer 2
It is still regrettable that the identification of all the volatile compounds was made without using authentic standard compounds, although most compounds detected in this study would be commercially available.The positive identification of volatiles (at least) related to the discrimination of jasmine teas could reinforce the results of this study.
Thank you for your understanding and encouragement! We will combine multiple methods for quantification and quantification in our future researches.
Firstly, as shown in figure 2A, the model parameters (R2Y= 0.966 and Q2 = 0.979) show that it has a high explained variance (R2Y) and cross-validation predictive capability (Q2). Then Fig. 2B shows the results of cross-validation, and the low intercepts (R2 = 0.437, Q2 = -0.661) show no over-fitting in the model.
Secondly, PC2 explains the second largest change in space. And it is intuitively obvious that there is no intersection existing in their projections on the Y axis (PC2).
In figure 5, samples cannot be discriminated according to their grades in group I.
Yes, as you indicated, samples in group I could not be discriminated according to their grades. That was why we believe E-nose could be good at distinguishing the grade difference caused by VOC concentration but deficient in identifying essential aromas that attribute to unique characteristics of high-grade jasmine tea. Our explaintions were shown in Line 330-340.
Line 73, compose? Is that ‘composition’?Corrected accordingly (Line 73). Thank you!
The nomenclatures of chemicals would not need to start with capital letters For example, ‘Methyl salicylate’, ‘Linalool’, ‘Linalool oxide, line 83-84.Rechecked and corrected accordingly. Thank you!
Line 95, ‘weight’should be replaced by ‘weighted’.Corrected accordingly (Line 95). Thank you!
Line 104-105, line 106: Please, rewrite the sentences.Corrected accordingly. Thank you!
“The primary thermal desorption of sampling tube was carried out at 250 ℃ for 5 minutes. In order to subject trapped compounds into gas chromatograph, the cold trap was then heated rapidly from -25 ℃ to 300 ℃. The temperature of valve and transfer line were maintained at 200 ℃ during analysis. Then, whole system was baken at 300 ℃ for 3 minutes as preparation of next sample analysis. ” (Lines 104-108)
Line 128-129 : Please, rewrite the sentences.Corrected accordingly. Thank you!
“A portion of sample (3.0 g) was weighed into a headspace bottle (60 ml) and equilibrated in a 55 ℃ water bath for 40 minutes.” (Lines 131-132)
Line 119, ‘Retention index’ needs no ‘capital letter’.Corrected accordingly (Line 121). Thank you!
Table 1, No. 4 and No 5. have the same RI values although they showed the differentWe have rechecked and modified it accordingly. Thank you!
Also, the Greek alphabets should be used for ‘alpha’, ‘beta’ and so on throughout the manuscript including tables.Corrected accordingly. Thank you!
What do you mean the Experimental m/z?‘Experimental m/z’ has been modified to ‘MS2 fragment’.
Please, confirm the nomenclature of ‘g-Cadinene’.Corrected to ‘γ-Cadinene’. Thank you!
How did you obtain the relative contents at the concentrations of x10 ug/g?In our experiment, the peak area of each identified VOC component was subjected to compared with the peak area of internal standard, through which the relative content of other VOC could be gave (Line 94-96).
【Wang C, Zhang C, Kong Y, et al. A comparative study of volatile components in Dianhong teas from fresh leaves of four tea cultivars by using chromatography-mass spectrometry, multivariate data analysis, and descriptive sensory analysis. Food Res Int. 2017;100(June):267-275. doi:10.1016/j.foodres.2017.07.013】
【Ding X, Wu C, Huang J, Zhou R. Characterization of interphase volatile compounds in Chinese Luzhou-flavor liquor fermentation cellar analyzed by head space-solid phase micro extraction coupled with gas chromatography mass spectrometry (HS-SPME/GC/MS). LWT - Food Sci Technol. 2016;66:124-133. doi:10.1016/j.lwt.2015.10.024】
Line 222, ‘correlational’ should be replaced by ‘correlationally’.Corrected accordingly (Line 224). Thank you!
Table1, 5 would be (E)-Linalool oxide.Corrected accordingly. Thank you!
Table S1 and S2 need references.Relevant references have been added ( Line 331, 130).
Reviewer 3 Report
The paper has been well reviewed and can be pubblished.
Author Response
Thank you again for your valuable suggestions and encouragement. I hope to learn more from you.